# Protocol for a multinational risk-stratified randomised controlled trial in paediatric Crohn's disease: methotrexate versus azathioprine or adalimumab for maintaining remission in patients at low or high risk for aggressive disease course

Rachel E Harris [ID],[1] Marina Aloi,[2] Lissy de Ridder [ID],[3] Nicholas M Croft,[4] Sibylle Koletzko,[5,6] Arie Levine,[7] Dan Turner,[8] Gigi Veereman,[9,10] Mattias Neyt,[11] Laetitia Bigot,[12] Frank M Ruemmele,[13,14] Richard K Russell [ID],[1] On behalf of PIBD SETQuality consortium and PIBDnet

FMR and RKR are joint senior authors.

For numbered affiliations see end of article.

**Correspondence to**
Dr Lissy de Ridder;
l.deridder@erasmusmc.nl

## ABSTRACT

**Introduction** Immunomodulators such as thiopurines (azathioprine (AZA)/6-mercaptopurine (6MP)), methotrexate (MTX) and biologics such as adalimumab (ADA) are well established for maintenance of remission within paediatric Crohn's disease (CD). It remains unclear, however, which maintenance medication should be used first line in specific patient groups.

**Aims** To compare the efficacy of maintenance therapies in newly diagnosed CD based on stratification into high and low-risk groups for severe CD evolution; MTX versus AZA/6MP in low-risk and MTX versus ADA in high-risk patients. Primary end point: sustained remission at 12 months (weighted paediatric CD activity index ≤12.5 and C reactive protein ≤1.5 fold upper limit) without relapse or ongoing requirement for exclusive enteral nutrition (EEN)/steroids 12 weeks after treatment initiation.

**Methods and analysis** REDUCE-RISK in CD is an international multicentre open-label prospective randomised controlled trial funded by EU within the Horizon2020 framework (grant number 668023). Eligible patients (aged 6–17 years, new-onset disease receiving steroids or EEN for induction of remission for luminal ± perianal CD are stratified into low and high-risk groups based on phenotype and response to induction therapy. Participants are randomised to one of two treatment arms within their risk group: low-risk patients to weekly subcutaneous MTX or daily oral AZA/6MP, and high-risk patients to weekly subcutaneous MTX or fortnightly ADA. Patients are followed up for 12 months at prespecified intervals. Electronic case report forms are completed prospectively. The study aims to recruit 312 participants (176 low risk; 136 high risk).

**Ethics and dissemination** ClinicalTrials.gov Identifier: (NCT02852694), authorisation and approval from local ethics committees have been obtained prior to recruitment. Individual informed consent will be obtained prior to participation in the study. Results will be published in a peer-reviewed journal with open access.

**Trial registration number** NCT02852694; Pre-results.

## Strengths and limitations of this study

► This is the first international prospective randomised controlled trial comparing three different medications for maintenance of remission in newly diagnosed Crohn's disease.
► This study may better define the most appropriate first-line immunomodulators based on a risk stratification protocol.
► Therapeutic efficacy will be supported by drug levels, pharmacogenomics and microbiome analysis as secondary outcomes.
► Inability to blind participants or treating physicians serves as a limitation to this study.
► Blinding of an alternative clinician to assess disease activity during study visits may prove practically difficult in smaller centres.

## INTRODUCTION

Crohn's disease (CD), the most common form of inflammatory bowel disease (IBD) in children, is a chronic disorder with the potential to affect the whole gastrointestinal tract. The aim of CD treatment is to control active inflammation and achieve bowel healing. Chronic and uncontrolled CD results in poor outcomes for patients, including reduced quality of life, recurrent hospitalisation and potential need for surgical intervention.[1]

Treatments for CD are categorised into those which induce remission (such as steroids[1 2] or exclusive enteral nutrition (EEN)[1 3] and those which maintain remission. Immunomodulators are a mainstay of maintenance treatment in IBD, with the efficacy of thiopurines (eg, azathioprine (AZA) and 6-mercaptopurine (6MP))[4–6] and methotrexate (MTX)[7–10] well established. Antitumour necrosis factor (anti-TNF) therapies (infliximab[11 12] and adalimumab (ADA)[13 14] including their biosimilars were used in those patients refractory to 'traditional' induction or maintenance treatment. More recently in clinical practice, patients deemed as high risk have been treated with a biologic without the need for prior use of an immunomodulator.

Due to a lack of treatment strategy trials within the paediatric IBD (PIBD) population, however, it remains unclear which of the aforementioned maintenance therapies should be used first line in individual patients. Randomised controlled trials (RCTs) comparing the use of MTX with thiopurines for maintenance of remission failed to show a significant difference in efficacy between the two.[15–17] A Cochrane review in adults with quiescent CD highlighted the lack of adequately powered trials necessary in order to determine the efficacy and safety of thiopurines compared with other maintenance therapies.[4 10] The RISK study (observational, non-randomised study) demonstrated improved clinical and growth-based outcomes at 1 year with anti-TNF monotherapy in comparison with immunomodulators; however, further investigation into which specific patients are most likely to benefit from these therapies is still required.[18] There is a clear disparity between North America and Europe in terms of which form of immunosuppression is used initially with both concerns about efficacy and safety lying behind these differences, thus, there is an urgent need for a head to head study in children to help objectively inform the primary choice of immunosuppression.

Stratifying patients by risk for complex or severe CD may allow pre-emptive direction of maintenance strategy and potentially an early reduction in disease burden with subsequent improvement in long-term outcomes. The adult IBD Ahead initiative highlighted young age at diagnosis as a risk factor for severity of CD evolution[19]; all patients diagnosed within paediatric services would therefore be considered 'high risk'. Paediatric consensus guidelines suggest that paediatric CD patients at 'high risk for poor outcome' should receive early therapy optimisation to modify progression of their disease.[1] The guidelines list specific features which may be considered predictive for poor outcome in paediatric CD (see table 1).[1] Patients deemed at high risk for complex disease or poor outcome may benefit from a 'top-down' approach as the TISKids (an RCT from disease diagnosis) aims to investigate.[20]

Therefore, the PIBDnet consortium recognised the urgent need to investigate the efficacy and safety of immunomodulators and to investigate whether a top-down approach was superior to a traditional 'step-up' for paediatric patients deemed at high risk for rapidly complicated disease course. REDUCE-RISK in CD is an RCT which aims to compare the effectiveness of immunomodulators for maintenance of remission in newly diagnosed CD based on risk stratification specifically, the effectiveness of MTX versus AZA/6MP for maintenance of remission who are low risk for rapidly progressive disease and the effectiveness of MTX versus ADA in a high-risk group.

## METHODS AND ANALYSIS
### Study design
We designed an international multicentre open-label prospective RCT with four treatment arms as shown

**Table 1** Definition of high-risk patients based on ECCO/ESPGHAN consensus guidelines

| Defining high-risk Crohn's disease patients | |
|---|---|
| **ECCO/ESPGHAN consensus guidelines** | **Modified study criteria** |
| Severe perianal disease | Complex perianal fistulising disease phenotype |
| Extensive (pan-enteric) disease; deep colonic ulcers on endoscopy | Panenteric disease phenotype (defined as L3 with L4b as per Paris classification[25] or L3 with deep ulcers in the duodenum, stomach or oesophagus not related to non-steroidal anti-inflammatory medications or *Helicobacter pylori*) |
| | Overall cumulative disease extent of $\geq 60\,cm$ |
| Stricturing and penetrating disease at onset | B2, B3 or B2B3 disease behaviour[20] |
| Marked growth retardation ≥2.5 height Z scores | Severe growth impairment (height z-score ≤2 or crossing ≥2 centiles) likely related to Crohn's disease |
| Persistent severe disease despite adequate induction therapy | Hypoalbuminaemia (<30 g/L), elevated CRP (at least two times upper limit of normal range), or wPCDAI >12.5 despite at least 3 weeks of optimised induction therapy with steroids or EEN |
| Severe osteoporosis | Not included |

CRP, C reactive protein; ECCO, European Crohn's and Colitis Organisation; EEN, exclusive enteral nutrition; ESPGHAN, European Society for Paediatric Gastroenterology, Hepatology and Nutrition; wPCDAI, weighted Paediatric Crohn's Disease Activity Index.

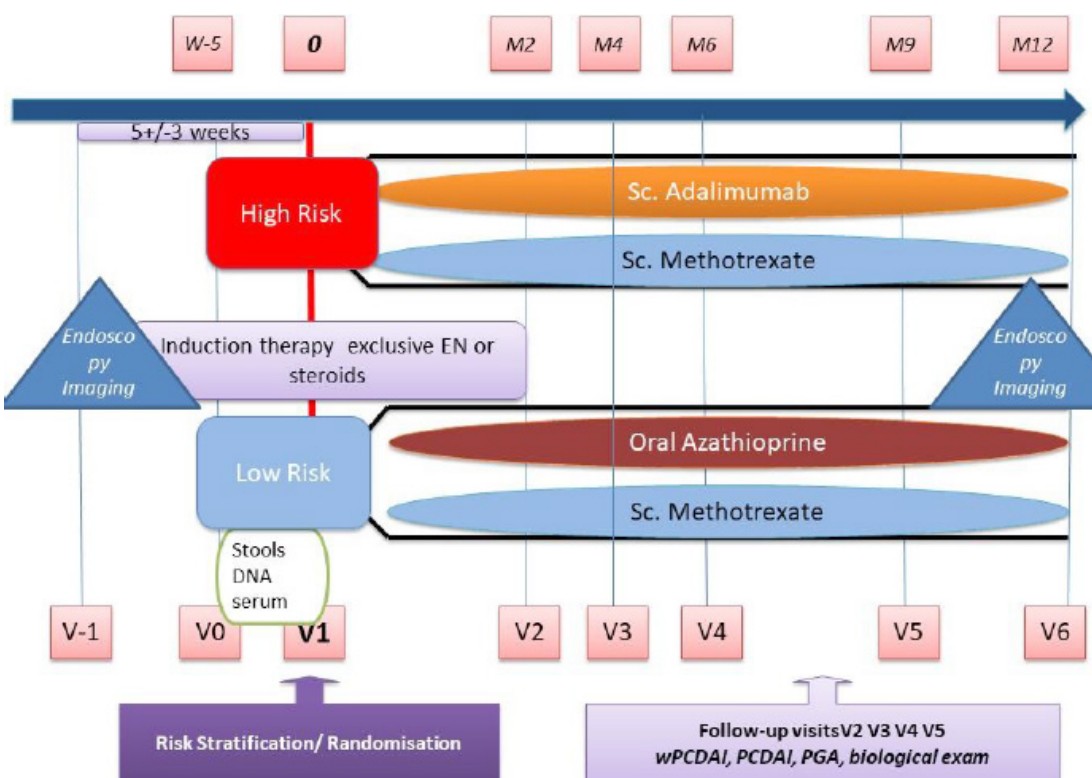

**Figure 1** Study design of the REDUCE-RISK in CD trial. M2=month 2, V2=visit 2. EN, enteral nutrition; PCDAI, Paediatric Crohn's Disease Activity Index; PGA, physician global assessment; wPCDAI, weighted PCDAI.

in figure 1. Following screening and consent, eligible patients are stratified into low and high-risk groups based on phenotype and disease response to induction therapy (table 1). Patients are then randomised to one of two arms within their risk group, with low-risk patients receiving either weekly subcutaneous MTX or daily oral AZA/6MP and high-risk patients receiving either weekly subcutaneous MTX or fortnightly subcutaneous ADA.

### Study end points

Patients are followed up for 12 months postrandomisation. The primary end point of the study is sustained steroid or EEN-free remission at 12 months, defined as weighted Paediatric Crohn's Disease Activity Index (wPCDAI) ≤12.5 and C reactive protein (CRP) ≤1.5 fold upper limit without a relapse or need for EEN/steroids since week 12.

Secondary end points include comparison of time to first relapse, remission at 12 weeks, growth, adverse events, health-related quality of life and patient-reported outcomes between the two treatment arms within each risk group but also between low and high risk MTX treated patients (a full list of secondary endpoints can be found in box 1). The TUMMY CD (patient-related outcome measure) was originally included as a secondary end point but has been withdrawn as the original timetable of development and validation of the score has not been met so it was not ready in time to be included. The study also aimed to evaluate clinical predictors for response, including genomic and serological markers and results of drug monitoring (MTX

and ADA concentrations) metabolites (6-thioguanine (6-TG) and 6-methylmercaptopurine (6-MMP) in AZA/6MP) and antidrug antibodies (ADA) in relation to adherence, toxicity and response. The ancillary study additionally aimed to evaluate the efficacy of ADA in patients treated from inclusion (top-down) versus patients switched to ADA due to immunomodulator failure (step-up). Further outcome measures are detailed in box 1.

### Eligibility criteria and recruitment

Full eligibility criteria for the study are listed in box 2. Patients are eligible if aged 6–17 years with new-onset (<6 months) treatment naïve luminally active and/or perianal fistulising CD diagnosed as per revised Porto criteria[21] receiving steroids or EEN for induction of remission with wPCDAI >40 or CRP >2 times upper limit of normal at diagnosis. Eligible definitions of disease behaviour were derived from the Paris classification.[22] Informed consent from must be obtained prior to participation in the study. Patients are excluded in cases of previous use of IBD-related medications, pregnancy or refusal to use contraceptives; disease requiring surgery, contraindications to study medication, exposure to live vaccine within 3 weeks, oral anticoagulant or antimalarial use, current or previous malignancy, significant infection or significant comorbidity.

The planned start date for the study is January 2017 with planned end June 2022.

## Box 1 Study endpoints

Primary endpoint
► Sustained steroid/exclusive enteral nutrition (EEN)-free remission at month 12, where sustained remission is defined as weighted Paediatric Crohn's Disease Activity Index (wPCDAI) ≤12.5 and C reactive protein (CRP) ≤1.5 times the upper limit without a relapse or need for EEN/steroids since week 12.

Secondary endpoints
Comparing the following within (1) the two treatment arms per risk group; (2) methotrexate treatment between high and low-risk groups and (3) tOP-down adalimumab (high-risk group) versus step-pp adalimumab (ancillary study):
► Rate of clinical remission at month 12 (physician global assessment, wPCDAI, PCDAI).
► Relapse free remission with normal CRP at month 12.
► Relapse-free remission with normal CRP and faecal calprotectin <300 at month 12.
► Remission at week 12 (measured by wPCDAI ≤12.5 and normal CRP and being off steroids/EEN).
► Time to first relapse after week 12.
► Predictive value of faecal calprotectin values at visits 1, 2, 4 and 6 (respectively at month 0, 2, 6 and 12).
► Drop-out rates.
► Adverse drug event rate including pharmacogenomics for toxicity and response to therapy.
► Height velocity and z-score at baseline and 52 weeks.
► Quality of life as measured by the IMPACT 3 questionnaire completed at each study visit.
► Health economic evaluation at all visits (forms EQ-5D-Y proxy 1, EQ-5D-Y and EQ-5D-5L, Productivity and Activity Impairment Questionnaire (WPAI):Crohn's disease Caregiver, School Attendance start of the research and follow-up visits).

## Box 2 Eligibility criteria

Inclusion criteria
► Patients aged 6–17 years with new-onset (<6 months) treatment naïve active luminal and/or perianal fistulising Crohn's disease diagnosed using established criteria[21] requiring steroids or exclusive enteral nutrition (EEN) for induction of remission weighted Paediatric Crohn's Disease Activity Index (wPCDAI) >40 or C reactive protein (CRP) >2 times upper limit of normal at diagnosis.
► Luminal active Crohn's disease (B1) with or without B2 and/or B3 disease behaviour as per Paris classification.[22]
► Signed informed consent.

Exclusion criteria
► wPCDAI <42.5 at diagnosis, except where CRP >2 times upper normal limit.
► Lack of induction therapy with steroids or EEN.
► Previous therapy with any inflammatory bowel disease-related medication other than induction therapy as detailed within this protocol with the exception of 5-aminosalicylic acid preparations.
► Pregnancy or refusal to use contraceptives during the study period in pubertal patients unless absolute abstinence is confirmed at each study visit.
► Lactating mothers.
► Perianal fistulising disease requiring surgical therapy.
► Patients homozygous for thiopurine methyltransferase (TPMT) mutations or those with TPMT activity <6 nmol/hour/mL erythrocytes or <9 nmol 6-methyl thioguanine (6MTG)/g Hb/hour, unless they qualify as high-risk patients.
► Evidence of undrained and uncontrolled abscess/phlegmon.
► Contraindication to any drugs used in the trial (including intolerance/hypersensitivity or allergy to study drugs (thiopurines, methotrexate or adalimumab)).
► Current or previous malignancy.
► Serious comorbidities (eg, renal insufficiency, hepatitis, respiratory insufficiency) which may interfere with drug therapy or interpretation of outcome parameters or will make it unlikely that the patient will complete the trial.
► Infection with mycobacterium tuberculosis, hepatitis B or C, HIV.
► Moderate to severe heart failure (New York Heart Association class III/IV).
► Oral anticoagulant therapy, antimalarial therapy.
► Live vaccine exposure (including yellow fever) less than 3 weeks prior to inclusion.

### Screening visit (visit 0)

The screening visit allows for assessment of eligibility for inclusion in the study, evaluation of the patient's response to induction therapy if already commenced, commencement of induction therapy where not commenced, and acquisition of consent and assent.

### Induction therapy

All enrolled patients receive either corticosteroids or EEN as induction as determined by the clinical team and the patient/caregiver. For EEN any balanced formula (polymeric or elemental) administered orally or via nasogastric tube is permitted and should be prescribed for 6–8 weeks. Tapering of steroids is at the discretion of the prescribing clinician. Adaptation of induction therapy (eg, dose increase of steroids or return to EEN) or crossover from one induction therapy to the other is permitted in order to achieve remission, however, patients must have discontinued their induction therapy by week 12. If induction therapy is not discontinued by week 12, the patient is considered a treatment failure, with the protocol for this detailed below.

### Inclusion visit and risk group allocation (week 5±3 weeks; visit 1)

In order to incorporate response to initial induction therapy within the risk stratification criteria, inclusion and risk group allocation is performed at week 5±3 weeks of induction therapy. Data from the screening visit are reviewed with ineligible patients excluded and patients are then stratified into the high or low risk group (table 1) based on the European Crohn's and Colitis Organisation/European Society for Paediatric Gastroenterology, Hepatology and Nutrition consensus guidelines.[1] Patients with perianal fistulising disease at diagnosis are auto allocated to the high-risk group regardless of other factors at inclusion visit. All other patients are allocated to the low-risk group. Patients with low thiopurine methyltransferase (TPMT) activity or homozygous mutations are excluded should they be categorised as low risk.

### Randomisation and treatment allocation

Randomisation is undertaken following allocation to high or low-risk group at week 5±3 weeks. This process uses an

**Table 2** Medication protocol for low and high-risk patients following randomisation

|  | Therapy | Route | Dose | Notes |
|---|---|---|---|---|
| Low risk protocol | Methotrexate | SC | 15 mg/m$^2$ body surface area weekly (max dose 25 mg). | Ondansetron 4–8 mg orally 1-hour preinjection and folic acid 15 mg (5 mg in patients <20 kg) 3 days postinjection are recommended for all patients. |
|  | **Versus** |  |  |  |
|  | Azathioprine | PO | 2.5 mg/kg (rounded down to nearest 12.5 mg). | Half calculated dose for TPMT heterozygotes/activity 6–9 nmol/hour/mL. |
|  | **OR** |  |  |  |
|  | 6-Mercaptopurine | PO | 1.5 mg/kg (rounded down to nearest 12.5 mg). | Half calculated dose for TPMT heterozygotes/activity 6–9 nmol/hour/mL. |
| High-risk protocol | Methotrexate | SC | 15 mg/m$^2$ body surface area weekly (max dose 25 mg). | Ondansetron 4–8 mg orally 1-hour preinjection and folic acid 15 mg (5 mg in patients <20 kg) 3 days postinjection are recommended for all patients. |
|  | **Versus** |  |  |  |
|  | Adalimumab (Humira) | SC | 160 mg then 80 mg after 2 weeks then 40 mg every 2 weeks thereafter (patients >35 kg). 120 mg then 80 mg after 2 weeks then 40 mg every 2 weeks thereafter (patients 25–35 kg). 80 mg then 40 mg after 2 weeks and 20 mg every 2 weeks thereafter (patients <25 kg). |  |

PO, Oral; SC, subcutaneous; TPMT, thiopurine methytransferase.

integrated module within the electronic case report form (CRF) system. Within both the high and low-risk groups patients are 1:1 randomised to MTX vs ADA or AZA/6MP, respectively, in blocks of four stratified by EEN or steroid induction therapy. Code for randomisation is prepared and held by the central coordinating site and site coordinators are then informed of the results. Immunomodulator or biological therapy should be commenced within 2 weeks of randomisation as per the protocol outlined in table 2.

AZA/6MP and MTX are prescribed and dispensed according to local guidelines. ADA (Humira) is provided by AbbVie. Cointerventions are prohibited.

### Follow up visits (visit 2, 3, 4, 5 and 6)
Patients are followed up at prespecified intervals (figure 1) with a window of ±2 weeks. A telephone call is undertaken at week 4 following initiation of induction in order to support patient compliance with induction regime and advise weaning where appropriate. Data as described in box 3 are collected at each consultation. Patients' compliance with therapy is determined at each face-to-face follow-up visit by pill and vial counts plus by patients' reporting.

Remission is defined as wPCDAI≤12.5, normal CRP (≤1.5 times upper normal range) and being free of steroids or EEN. Once remission is achieved and induction therapy is discontinued, a patient is considered to be failing treatment or experiencing a relapse in the following circumstances:

► wPCDAI >40.
► CRP >2 times upper normal limit in the absence of any clear infectious process.
► wPCDAI >12.5 but <40 and/or CRP >1.5 times but <2 times over upper normal limit at two consecutive visits within 2–8 weeks.
► Development of CD-related complications, for example, fistulisation.
► Requirement for additional CD-specific medication/ surgery since last study visit.

A patient will also be considered a treatment failure should induction therapy be continued at week 12. In addition, the treating clinician may escalate treatment at any time point independent of wPCDAI score if it is felt that the patient is experiencing a relapse.

### Dose optimisation and therapeutic drug monitoring
Drug monitoring is undertaken as detailed below. In addition to this, samples for drug monitoring should be collected at the time of medication cessation in the event of drug discontinuance due to adverse effect or loss of response. Potential adaptations to therapies which may be made at specific follow-up visits are detailed in box 4.

## Box 3   Standard requirements for each study visit

► An explicit history of illness since last visit, including review of symptoms, medications (including compliance check) and adverse events.
► Physical examination weighted Paediatric Crohn's Disease Activity Index (wPCDAI), physician global assessment and PCDAI scoring.
► Anthropometrics (height measured using a calibrated wall mounted stadiometer).
► Blood tests.
   – White blood cells.
   – Absolute neutrophil count.
   – Haemoglobin.
   – Haematocrit.
   – Platelet count.
   – Erythrocyte sedimentation rate.
   – C reactive protein.
   – Amylase.
   – Albumin.
   – Aspartate transaminase.
   – Alanine transaminase.
   – Conjugated bilirubin.
   – Gamma glutamyl transferase.
► Stool samples for faecal calprotectin and microbiome analysis.
► Health economic parameters (EQ-5D-Y proxy 1; EQ-5D-Y; EQ-5D-5L; WPAI:Crohn's disease; school attendance questionnaire).
► Quality of life evaluation (IMPACT 3).
► Urine human chorionic gonadotropin in all female patients of child-bearing potential.
► Confirmation of contraception use or of absolute abstinence in all patients.

## Box 4   Potential adaptations to therapies at follow-up visits

Month 2 (visit 2)
► Failure to discontinue induction therapy by week 12
   – Offer switch to the ancillary study (adalimumab (ADA) step-up) to those prescribed methotrexate (MTX) or azathioprine (AZA)/6-mercaptopurine (6MP), or an increase in dose frequency to weekly in those prescribed ADA.
   – Alternatively, the patient may leave the study and receive therapies as per the discretion of the treating clinician.
Months 4, 6, 9 and 12 (visits 3, 4, 5 and 6)
► Thiopurine non-response.
   – Protocol as per metabolite levels (detailed in table 3).
► Thiopurine intolerance (except pancreatitis).
   – Switch to alternate thiopurine (AZA to 6MP or vice versa) or split dose to provide twice daily (BD) dosing.
► Thiopurine failure (any exacerbation despite dose optimisation/pancreatitis/cytopaenia).
   – Offer switch to ancillary study (ADA STEP-up) or exit study.
► MTX intolerance or failure (any exacerbation or elevation of liver enzymes as detailed below).
   – Offer switch to ancillary study (ADA STEP-up) or exit study.
► ADA failure (any exacerbation).
   – Increase frequency to weekly dosing.

**Table 3**   Azathioprine dose adjustments based on metabolite levels

| Result | Action |
| --- | --- |
| 6-TG <150 | Consider non-compliance; repeat sample at subsequent visit and increase dose if low 6-TG confirmed (+25 mg or +12.5 mg if dose <50 mg). |
| 6-TG 150–800 | No adaptation. |
| 6-TG >800 | Decrease dose if repeat sample at subsequent visit confirms high 6-TG (−25 mg or −12.5 mg if dose <50 mg). |
| 6-MMP >8000 or signs of hepatotoxicity | Stop medication—switch to ancillary study. Erythrocyte lysate sample frozen at −80°C and shipped to central lab at end of study for thiopurine nucleotides. |

6-MMP, 6-methylmercaptopurine.

### Azathioprine

TPMT genotype or phenotype at screening determines the initial dose of AZA/6MP; and measurement of thiopurine metabolites (6-TG and 6-MMP) at visit 2 determines requirement for subsequent dose adjustment performed according to the recommendations in table 3. Where possible thiopurine metabolites are measured locally; central lab measurements are provided for centres where this is unavailable.

At visit 2, a urine sample for TPMT metabolite determination and an erythrocyte lysate sample for quantification of thiopurine nucleotides by liquid chromatography-tandem mass spectrometry should be frozen at −80°C and shipped on dried ice to the central lab at the end of the study. At each visit from visit 2–6, an additional ethylenediaminetetraacetic acid (EDTA) blood sample will be collected for further 6-TG and 6-MMP testing and frozen at −80°C.

### Methotrexate

Washed erythrocyte for MTX levels will be obtained at visits 2, 4 and 6 and stored frozen at local centres. These samples will be sent on dry ice for central analysis to evaluate response to therapy and adverse effects in relation to drug levels.

### Adalimumab

ADA trough levels are measured after three injections of maintenance therapy (eg, at visit 2) within the local laboratory (central lab testing available if local lab testing is unavailable). Dosing interval may be shortened to weekly in the event of low ADA levels (<8 µg/mL) and negative ADA antibodies. Further samples should be obtained at visits 3, 4, 5 and 6 and should be frozen for later analysis within the central lab.

## Pharmacogenetics

DNA for pharmacogenetics should be taken from patients randomised to MTX or AZA/6MP for multiplex genotyping of polymorphism related to drug metabolism to evaluate safety and response to therapy. Analysis will be performed at the end of the study, or earlier in those patients showing toxicity.

## Ancillary study

Patients unable to discontinue induction therapy or those randomised to thiopurine or MTX therapy who experience treatment failure may be invited to participate in the ancillary study (STEP-up ADA) until visit 5. Any initial maintenance therapy will be stopped and induction and maintenance regime for ADA as previously described will be commenced. Up to three additional study visits at 3-month intervals will be offered to these patients in order to obtain 12 months of follow-up. A maximum of 68 patients can participate in this ancillary study allowing a 1:1 comparison of top-down ADA to step-up ADA therapy.

## Unscheduled visits

Unscheduled visits may be arranged based on clinical requirements. As for scheduled visits per protocol treatment adaptions are possible if intolerance or failure of the study drug is detected. Subsequent scheduled visits will not be changed after an unscheduled visit.

## Treatment discontinuation

Patients who discontinue treatment before completing 12 months of study drug within either the main study or the ancillary study will receive a single follow-up visit. This will be either 12 months after the commencement of study treatment or at the point of inclusion in the ancillary study.

Modifications to the protocol while the study is being conducted will be relayed to all site staff by email and then onto their relevant ethical and regulatory boards. The current manuscript is based on protocol 5.1 last modified 28 May 2019.

## Allocation concealment and blinding

For ethical reasons, we decided against a double dummy design for blinding the patient, parents and care givers. Due to the differences in medication administration route and the significant nausea commonly associated with MTX blinding of the allocation to the patients, their families or their physicians is not possible. Where possible, however, blinding of an alternative clinician to score the wPCDAI, PCDAI and physician global assessment at each study visit should occur (prospective randomised open blind end-point (PROBE) evaluation).

## Safety

The external and independent advisory board of PIBD-SETQuality serves as an independent data and safety monitoring board which meets at prespecified intervals with access to all data within the study. The principal investigator at each site is responsible for reporting any safety issues (adverse events, serious adverse events (SAEs), suspected

> **Box 5 Criteria for premature termination of study treatment or participation**
>
> ► Pregnancy at any stage.
> ► Treatment failure as per protocol.
> ► Failure to tolerate allocated treatment or alternatives as listed within the protocol.
> ► Significant drug-related side effects manifesting as significantly abnormal bloods results or adverse effects based on the clinical judgement of the treating physician.
> ► Request of participant to be withdrawn from treatment.
> ► The judgement of the treating physician being that it is in the best interests of the participant to withdraw from study treatment.
> ► Loss of participant to follow-up.
> ► Patient death.

unexpected serious adverse reactions), drop-outs or any new information which may impact the study in any way. The principal investigator shall report to the sponsor all SAEs experienced by a study subject receiving an ADA (Humira) within 24 hours of learning of the event regardless of the relationship of the event to the product. All SAEs are immediately sent to AbbVie pharmacovigilance by the sponsor. SAEs will be followed from the date of patient's signature of informed consent, until complete resolution or 30 days after the end of the study/patient's final study visit.

Participants may withdraw consent for further participation or data collection at any time without giving reason and without prejudicing further care or treatment. Patients will be permanently withdrawn from study treatment in the event of any of the situations outlined in box 5. Patients should be provided with a study alert card for use in the event of an emergency.

Biochemical markers are monitored with a clearly defined protocol for adjustments to therapy based on abnormal results (eg, neutropenia, pancreatitis, elevated liver enzymes).

## Data collection, management and monitoring

Patient CRFs are completed in a prospective manner using an electronic web-based system designed specifically by PIBDnet for this trial. In order to maintain data security and integrity, the web-based data entry will be linked to a password-secured Microsoft Access database, where data will be stored until time of analysis. Files will be saved on a code secured net-drive and backed-up following each data entry on a disk locked in a cabinet. Patients will be identified only by a study code assigned at the point of enrolment. Code of patient identifiers will be kept at each participating site. Handling of patient-identifiable is compliant with the legislation of each participating centre and the European General Data Protection Regulation. Investigators will be invited to fax or email the paper source document to the coordinating site on a random basis to allow appropriate monitoring. Access to data with detailed information on study outcomes will be made available to other research

groups on request and at the discretion of the principal investigators.

Monitoring arrangements are in place for all sites after initial site initiation. The monitoring visits will occur regularly partly dependant on recruitment rate at individual sites. The monitoring is performed usually by someone external to the clinical team.

### Analysis and statistical methods

Descriptive statistics (mean, median, SD, SE, quartiles, minimum, maximum and two-sided 95% confidence limits of mean and median) will be presented for each treatment of the low and high-risk paediatric CD groups and where applicable, for the paired difference of each patient. Frequency tables will be presented where applicable.

### Primary analysis

Difference in the 12-month steroid/EEN free sustained remission rates between the treatment groups will be undertaken using $X^2$ test. Mantel-Haenszel test will be used to combine data from all participating sites.

### Secondary analyses

$X^2$ tests or Fisher's exact tests will be used to compare rates of remission, steroid intake, drop-out and SAEs between the two arms of each risk group and between the low and high-risk MTX groups. Logistic regression analyses may be performed to adjust for any imbalances in baseline covariates. To compare time to disease flare between the arms of each risk group and between high and low-risk MTX groups, a Kaplan-Meier survival estimate will be used and the log-rank test of equality over strata. A Cox proportional hazard model will be constructed to obtain a HR after validation of the proportionality assumption and adjusting for possible confounding variables (including age and disease duration). Student's t tests or Wilcoxon rank sum tests will be used to compare growth, steroid dose, adverse events, changes in quality of life and patient reported outcomes between the two arms of each risk group and between the high and low-risk MTX groups. The predictive value of faecal calprotectin levels, CRP, serum tests or other clinical predictors for response (including genomic and serological markers) will be assessed for each arm of the study using sensitivity, specificity, negative and positive predictive values or area under the receiver operating characteristic curve. Multivariate logistic regression analyses will then be performed.

Analyses will be performed using the R software (http://cran.r-project.org). All comparisons will be made using a two-sided significance level of 0.05.

### Sample size considerations

Estimated remission rates are based on recent analysis from the RISK study,[18] indicating an advantage of early anti-TNF introduction over immunomodulator therapy. For the low-risk group, it was hypothesised that 48% of children will be in remission at 12 months for the AZA/6MP arm versus 70% for the MTX arm. On the basis of this data with an alpha risk of 5% and a power of 80% a sample requirement of 88 patients per arm was calculated assuming a 10% lost to follow-up. For the high-risk group, it was hypothesised that 40% of children will be in remission at 12 months for the MTX arm versus 65% for the ADA arm. To detect this difference with an alpha risk of 5% and a power of 80%, a sample size of 68 participants is necessary, again assuming a 10% lost to follow-up. In total 312 participants will be included in the study (176 low-risk group; 136 high-risk group).

### Patient and public involvement

Patients were not involved in the development of this study; however, the French patient charity AFA Crohn, RCH, France was involved in study design and critically reviewed and commented on all aspects of the trial.

### Discussion

REDUCE-RISK in CD is the first multicentre international RCT aiming to compare three different medication strategies for maintenance of remission in newly diagnosed CD based on a risk stratification protocol. During the 12-month follow-up period, the effects of the differing management strategies will be assessed via data collected and outcome measures as defined above in order to analyse the efficacy and safety of each medication and better define the most appropriate first-line maintenance immunomodulators to be used in specific subsets of CD patients. As a group we speculatively hypothesise that MTX will be superior to thiopurines for maintaining remission in CD in the low-risk group although in the absence of head to head studies prior to this one this study will provide randomised data to address this. Additionally, from our own work and others, we know response to induction therapy is an important prognostic marker and we wanted to allow the induction treatment to have a chance to work before we assigned high or low risk status.[23 24] Thus, it was a pragmatic compromise with the timing of introduction of the maintenance treatment to give the induction treatment long enough to show its effect while recognising both treatments have a 'lag period' of a few weeks before they become fully effective.

We also hypothesise that ADA will be superior to MTX in the high-risk group based on the results from the RISK study.[18] Of note ADA (Humira) was chosen to allow delivery of the study out of hospital, to reduce drug costs and it allowed single therapies to be compared with each other. Practically if we had used infliximab (Remicade) then we would have needed to use combination therapy which we did do not want to do as it would have further complicated the trial design.

In addition to this, the ancillary study will compare outcomes in ADA treated patients from inclusion (top-down) versus patients switched to ADA due to failure of immunomodulators (step-up), with the potential to stratify which patients might benefit from such a top-down treatment strategy. We acknowledge that comparison of the ancillary group with the group randomised from baseline to ADA is not randomised and may be subject to selection bias noting the ancillary group have failed or been intolerant to initial therapy. However, we feel it is important to

include this to allow us to compare the trial with studies which have allocated patients directly to anti-TNF (RISK, TISKids) and to see how many patients benefit from 'rescue therapy' after failure of their initial allocation.

The design and completion of interventional studies in PIBD is a recognised challenge between rigorous study design methodology and pragmatic considerations around feasibility and completion within a paediatric dataset.[25] This particular study is limited by the inability to blind the treatment allocation to the patients, their families or their treating physician due to the differences in medication administration route and the side effects commonly associated with the study medication. Although the protocol advises that where possible blinding of an alternative clinician to score disease assessment at each study visit should occur in order to obtain PROBE evaluation, this may be practically difficult in smaller centres where staff are familiar with the majority of their patient cohort.

## ETHICS AND DISSEMINATION

The study is being conducted according to the principles of the Declarations of Helsinki and to date has been approved by all participating sites as listed within online supplementary table 1. Clinical trials authorisation and ethics approval has been obtained from the local ethics review committees of these participating nations and centres. The Standard Protocol Items: Recommendations for Interventional Trials guidelines[26] were adhered to in the production of the protocol for this trial (see uploaded material for details).

### Consent

Patients and their caregivers are provided with study-specific information including an explicit description of the study outline and alternatives for participation. It is made clear to all patients approached that declining to participate in the study will not jeopardise the quality of subsequent care received. After a period of consideration, if agreeable, the patient's parent or caregiver is asked to sign consent forms with age-appropriate assent obtained from the child where relevant (see online supplementary appendix 1 for model consent forms). The signed forms are filed within the patient's medical record with a copy provided to the participant and their caregiver. Consent will be obtained by site staff with the relevant training and who are identified as assigned on the delegation log. Participants taking part in the ancillary study will not be reconsented.

### Dissemination

Results of the study will be submitted for publication within a peer-reviewed journal. In accordance with the H2020 general grant agreement, the dissemination process will ensure open access to the scientific publications resulting from this project. Journal authorship guidelines will be adhered to and there are no plans to use professional writers.

**Author affiliations**
[1]Department of Paediatric Gastroenterology, Royal Hospital for Children Glasgow, Glasgow, UK

[2]Paediatric Gastroenterology and Liver Unit, Sapienza University of Rome, Roma, Lazio, Italy
[3]Paediatrics, Erasmus MC/Sophia Childrens Hospital, Rotterdam, The Netherlands
[4]Department of Paediatric Gastroenterology, Barts and The London School of Medicine and Dentistry, London, UK
[5]Pediatric Gastroenterology and Hepatology, Dr. V. Hauner Children's Hospital, Munich, Germany
[6]Department of Pediatrics, Collegium Medicum University of Warmia and Mazury, Olsztyn, Poland
[7]Edith Wolfson Medical Center, Tel Aviv University, Tel Aviv, Israel
[8]Department of Paediatric Gastroenterology, Hebrew University of Jerusalem, Jerusalem, Israel
[9]Pediatric GI, UZBrussels-VUB, Brussels, Belgium
[10]Free University Brussels, University Hospital, Brussels, Belgium
[11]ME-TA Medical Evaluation and Technology Assessment, Merendree, Belgium
[12]PIBD-Net, Hôpital universitaire Necker-Enfants malades, Paris, Île-de-France, France
[13]Service de Gastroentérologie Pédiatrique, Hôpital Universitaire Necker-Enfants Malades, Paris, Île-de-France, France
[14]Department of Paediatric Gastroenterology, Université Paris Descartes, Paris, Île-de-France, France

**Contributors** REH prepared the draft manuscript with comments and review from all authors. RKR, MA, LdR, NMC, SK, AL, DT, GV, MN and LB and FMR were involved in the conception, design, planning and then drafting of the original research protocol and RKR, REH, MA, LdR, NMC, SK, AL, DT, GV, MN and LB and FMR provided critical review of the manuscript and approved the final uploaded draft. As sponsor PIBDnet has full responsibility and control for the original study design, collection, management, analysis and interpretation of data, including writing of the report and the decision where to submit the report for publication.

**Funding** This work was supported as part of the PIBD-SETQuality (Paediatric Inflammatory Bowel Diseases Network for Safety, Efficacy, Treatment and Quality improvement of care) project funded by the European Commission Horizon 2020 - Research and Innovation Framework Programme (grant number 668023). ADA (Humira) is provided by AbbVie.The main study sponsor is PIBDNet. PIBDNet is the EU legal representative for the study. The specific contact for the sponsor is Frank Ruemmele (Service de Gastro-entéroloegie, Hôpital Necker Enfants Maldes, 149 rue de Sèvres, 75015 Paris, France).

**Competing interests** RKR is supported by an NHS Research Scotland Senior Research Fellowship, and has received speaker's fees, travel support and/or participated in medical board meetings with Nestle, MSD Immunology, AbbVie, Dr Falk, Takeda, Napp, Mead Johnson, Nutricia & 4D Pharma. FMR has received speaker fees from Shering-Plough, Nestlé, MeadJohnson, Ferring, MSD, Johnson & Johnson, Centocor, AbbVie; has served as a board member for SAC:DEVELOP(Johnson & Johnson), CAPE (AbbVie), LEA (AbbVie); and has been invited to MSD France, Nestlé Nutrition Institute, Nestlé Health Science, Danone, MeadJohnson, Takeda, Celgene, Biogen, Shire, Pfizer and Therakos. DT received consultation fee, research grant, royalties, or honorarium from Janssen, Pfizer, Hospital for Sick Children, Ferring, Abbvie, Takeda, Biogen, Atlantic Health, Shire, Celgene, Lilly, Neopharm, Roche. LdR received consultation fee, research grant, or honorarium from ZonMw, ECCO, Shire, Malinckrodt, Nestlé, Celltrion, Abbvie and Pfizer. MA received consultation fee and honorarium from Abbvie. SK received consultation fee, research grant, or honorarium from Danone, Nestec-Nutrition, Abbvie, Takeda, Celgene, Shire, Pfizer, Biogaia, Janssen, Berlin-Chemie; Mead Johnson, Vifor, Pharmacosmos, ThermoFisher

**Patient and public involvement** Patients and/or the public were not involved in the design, or conduct, or reporting, or dissemination plans of this research.

**Patient consent for publication** Not required.

**Provenance and peer review** Not commissioned; externally peer reviewed.

**ORCID iDs**
Rachel E Harris http://orcid.org/0000-0001-6507-3487

Lissy de Ridder http://orcid.org/0000-0002-6035-1182
Richard K Russell http://orcid.org/0000-0001-7398-4926

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
