## [Reviewer comments · BMJ Open]

ARTICLE DETAILS

TITLE (PROVISIONAL)	PROTOCOL FOR A MULTI-NATIONAL RISK-STRATIFIED RANDOMISED CONTROLLED TRIAL IN PAEDIATRIC CROHN'S DISEASE: METHOTREXATE VERSUS AZATHIOPRINE OR ADALIMUMAB FOR MAINTAINING REMISSION IN PATIENTS AT LOW OR HIGH RISK FOR AGGRESSIVE DISEASE COURSE
AUTHORS	Harris, Rachel E; Aloï, Marina; de Ridder, Lissy; Croft, Nicholas M; Koletzko, Sibylle; Levine, Arie; Turner, Dan; Veereman, Gigi; Neyt, Mattias; Bigot, Laetitia; Ruummele, Frank; russell, Richard K

VERSION 1 – REVIEW

REVIEWER	Viktor Wintzell Karolinska Institutet, Sweden
REVIEW RETURNED	11-Dec-2019

GENERAL COMMENTS	This is the protocol for an RCT, REDUCE-RISK in CD, that investigates the efficacy of common maintenance treatments (MTX, AZA, ADA) in a risk-stratified population of children with Crohn's disease. It is an international multicenter trial that aims to recruit 312 participants; patients with new-onset CD, age 6-17 years, receiving steroids or EEN as induction therapy. Given the lack of evidence on optimal first-line maintenance treatment this is an important and anticipated study, in particular the high risk group analysis. Find my comments below: For the ancillary analysis (STEP-up ADA), patients who not receive ADA on baseline and who lack treatment response are included. These patients are low and high risk patients randomized to MTX or AZA on baseline, as well as patients who were unable to discontinue induction therapy. They will be compared with high risk patients randomized to ADA at baseline (TOP-down ADA). Given the different risk levels at baseline and that the analysis is non-randomized (comparison is both between and within risk groups) the purpose of this analysis is not clear. Further, the inclusion in the ancillary analysis is conditioned on failure of initial maintenance therapy with immunomodulators, which makes the analysis susceptible to selection bias, when compared with patients randomized to ADA at baseline. If the purpose is to compare top-down (first ADA) and bottom-up (first MTX or AZA; then ADA if lack for response) treatment strategies it seems that the primary (randomized) analysis of the high risk group (ADA Vs MTX on baseline) is more appropriate. Please explain and elaborate on aim and potential bias. The overall aim of the study is to investigate the efficacy of first-line maintenance therapies in high and low risk patients (MTX Vs
---

	AZA; MTX Vs ADA). The high risk analysis could confirm the results of the RISK study (ref 18), which showed higher effectiveness of TNFi at baseline in comparison with immunomodulators (suggest to clarify that this is an observational study), in a high risk population. Does it also aim to identify subgroups with higher relative effectiveness of ADA? Were additional risk groups with ADA arm considered? Background to low risk group analysis (MTX Vs AZA) and hypothesized effects are unclear. Did the previous trials in adults (ref 15-17) indicate that MTX is more efficacious than AZA, despite small sample sizes? Please explain the basis for hypothesized proportions in remission at 12 months: 70% for MTX and 48% for AZA, which implies a ratio of 1.4. Recent meta-analysis (Colman 2018) of MTX in pediatric CD showed that 37% were in remission after 12 months based on 14 observational studies (886 patients). TNF inhibitor infliximab is common in pediatric CD. How does the current trial build on the evidence that has been accumulated with regards to infliximab? Immunomodulators and immunomodulatory medications refer interchangeably to MTX, thiopurines, ADA (e.g. introduction in abstract, introduction) and to MTX and thiopurines only (e.g. in article summary, section 'Randomisation and Treatment Allocation'). Please edit for consistency. Language, suggest to rephrase: P9, R31: 'to stratify' in sentence '... the urgent need to investigate the efficacy and safety of immunomodulatory medications and to stratify whether a topdown ...' P7, R52: 'first' in sentence '... in terms of which form of immunosuppression first with both concerns about efficacy and safety ...' Suggest to edit figure 1 to include ancillary analysis, use colors consistently, avoid line breaks, clarify meaning of M2-M12, V2-V6. What is the time plan for the study? Has enrollment begun and what are the tentative dates? Suggest to describe this, if not already included.
--	--

REVIEWER	Lara Hart McGill University, Montreal QC, Canada
REVIEW RETURNED	31-Dec-2019

GENERAL COMMENTS	Very interesting and relevant research question and methodology. Very clear proposal. A few questions: a) why did you choose to use Humira instead of Remicade for the biologic arm? b) Why did you choose to start maintenance therapy in the low risk group at 5+3 weeks into induction, when it can take 8+ weeks for MTX and imuran to work (thereby potentially causing patients to still need induction therapy > 12 weeks) c) what are the traditional/accepted MTX levels in the blood? (as implicated on pg 21)
--

	A few comments: Pg 7 - line 15-22: please revise to explain that biologics had previously (or traditionally) been reserved to those who failed immunomodulators (as, since the ECCO/ESPGHAN guidelines identified high risk features, many of us have been using biologics without previous IM if the patient presents with high risk features) pg 7 - line 49-56: I don't understand what the sentence is saying; please modify sentences/explain further Overall, please be careful of sentence structure and please add commas (as it otherwise makes it somewhat difficult to read)
--	---

VERSION 1 – AUTHOR RESPONSE

Reviewer: 1

Reviewer Name: Viktor Wintzelle

Institution and Country: Karolinska Institutet, Sweden

Please state any competing interests or state 'None declared': None declared

Please leave your comments for the authors below

This is the protocol for an RCT, REDUCE-RISK in CD, that investigates the efficacy of common maintenance treatments (MTX, AZA, ADA) in a risk-stratified population of children with Crohn's disease. It is an international multicenter trial that aims to recruit 312 participants; patients with new-onset CD, age 6-17 years, receiving steroids or EEN as induction therapy. Given the lack of evidence on optimal first-line maintenance treatment this is an important and anticipated study, in particular the high risk group analysis.

Response: Thank you very much for your kind comments and summary of the study.

Find my comments below:

For the ancillary analysis (STEP-up ADA), patients who not receive ADA on baseline and who lack treatment response are included. These patients are low and high risk patients randomized to MTX or AZA on baseline, as well as patients who were unable to discontinue induction therapy. They will be compared with high risk patients randomized to ADA at baseline (TOP-down ADA). Given the different risk levels at baseline and that the analysis is non-randomized (comparison is both between and within risk groups) the purpose of this analysis is not clear.

Response: Thanks the reviewer is correct this analysis is non randomised but is in place to provide an overall guide to see if more patients were allocated to ant-tnf from baseline what their approximate outline might have been. The RISK study and TISKids study have both tried this approach and having this additional arm will help inform future studies using early parameters for stratification how many would then benefit from "rescue treatment".

Further, the inclusion in the ancillary analysis is conditioned on failure of initial maintenance therapy with immunomodulators, which makes the analysis susceptible to selection bias, when compared with patients randomized to ADA at baseline. If the purpose is to compare top-down (first ADA) and bottom-up (first MTX or AZA; then ADA if lack for response) treatment strategies it seems that the primary (randomized) analysis of the high risk group (ADA Vs MTX on baseline) is more appropriate. Please explain and elaborate on aim and potential bias.

Response: Thanks. We acknowledge this and to deal with this and the previous point have added the following to the end of the first paragraph of the discussion.

“We acknowledge that comparison of the ancillary group with the group randomised from baseline to ADA is not randomised and may be subject to selection bias noting the ancillary group have failed or been intolerant to initial therapy. However we feel it is important to include this to allow us to compare the trial with studies which have allocated patients directly to anti-tnf (RISK, TISKids) and to see how many patients benefit from “rescue therapy” after failure of their initial allocation.”

The overall aim of the study is to investigate the efficacy of first-line maintenance therapies in high and low risk patients (MTX Vs AZA; MTX Vs ADA). The high risk analysis could confirm the results of the RISK study (ref 18), which showed higher effectiveness of TNFi at baseline in comparison with immunomodulators (suggest to clarify that this is an observational study), in a high risk population. Does it also aim to identify subgroups with higher relative effectiveness of ADA? Were additional risk groups with ADA arm considered?

Response: Thanks. We have clarified when first mentioned in the text that the RISK study is an observational non-randomised cohort and for consistency have therefore also added the TISKids study is an RCT.

AS the RISK study was observational this study aims to add to and improve on the analysis done by the RISK group by using a randomised allocation. We will use the detailed clinical and scientific data collected to look at predictors of ADA response but we did not add any additional arms for ADA for largely pragmatic reasons and to maintain feasibility of study outcome and completion

Background to low risk group analysis (MTX Vs AZA) and hypothesized effects are unclear. Did the previous trials in adults (ref 15-17) indicate that MTX is more efficacious than AZA, despite small sample sizes? Please explain the basis for hypothesized proportions in remission at 12 months: 70% for MTX and 48% for AZA, which implies a ratio of 1.4. Recent meta-analysis (Colman 2018) of MTX in pediatric CD showed that 37% were in remission after 12 months based on 14 observational studies (886 patients).

Response: Thanks for this. We used paediatric data much of it from authors centres and alot of it unpublished. This included data from the GROWTH study as this is a large European multicentre paediatric study to generate this pilot data. Although data from the study has been published the specific data on which we based these calculations has not. We generated the data for the study before the Cochrane review was published.

TNF inhibitor infliximab is common in pediatric CD. How does the current trial build on the evidence that has been accumulated with regards to infliximab?

Response: Thanks. This trial does not include infliximab which indeed is a common anti-tnf but in many places ADA is just as if not more commonly used. Therefore it will not add anything directly to the accumulated evidence for infliximab.

Immunomodulators and immunomodulatory medications refer interchangeably to MTX, thiopurines, ADA (e.g. introduction in abstract, introduction) and to MTX and thiopurines only (e.g. in article summary, section ‘Randomisation and Treatment Allocation’). Please edit for consistency.

Response: Thanks alot. Immunomodulators are the term now used throughout with replacement of immunomodulatory medicines where appropriate. Sorry for the confusion.

Language, suggest to rephrase: P9, R31: ‘to stratify’ in sentence ‘... the urgent need to investigate the efficacy and safety of immunomodulatory medications and to stratify whether a topdown ...

Response: thanks stratify has been changed to investigate in this sentence

' P7, R52: 'first' in sentence '... in terms of which form of immunosuppression first with both concerns about efficacy and safety ...'

Response: The sentence has been modified in several places including the one suggested by the reviewer and now reads as:

"There is a clear disparity between North America and Europe in terms of which form of immunosuppression is used initially with both concerns about efficacy and safety lying behind these differences, thus there is an urgent need for a head to head study in children to help inform the primary choice of immunosuppression."

Suggest to edit figure 1 to include ancillary analysis, use colors consistently, avoid line breaks, clarify meaning of M2-M12, V2-V6.

Response: Thanks this is a very valid point. Before we make any graphic changes we await editorial comment as in our experience the figures often have to be modified at the proofing stage so if accepted we will work with the journal to maximise the appearance of the figure then. We have added as suggested the meaning of M2, V2 in the meantime though.

What is the time plan for the study? Has enrollment begun and what are the tentative dates? Suggest to describe this, if not already included.

Response: Thanks the study has already begun. In responding to this point as already brought up by the editor too we have inserted the relevant dates into the methodology section.

Reviewer: 2

Please leave your comments for the authors below

Very interesting and relevant research question and methodology. Very clear proposal.

Response: Thank you

A few questions:

a) why did you choose to use Humira instead of Remicade for the biologic arm?

Response: Humira was chosen to allow delivery of the study out of hospital, to reduce drug costs and it allowed single therapies to be compared with each other. Practically if we had used Infliximab (Remicade) then we would have needed to use combination therapy which we did not want to do as it would have further complicated the trial design.

b) Why did you choose to start maintenance therapy in the low risk group at 5+3 weeks into induction, when it can take 8+ weeks for MTX and imuran to work (thereby potentially causing patients to still need induction therapy > 12 weeks)

Response: Thanks. From our own work and others we know response to induction therapy is an important prognostic marker and we wanted to allow the induction treatment to have a chance to work before we assigned high or low risk status. (Ziv-Baran T, Hussey S, Sladek M, et al. Response to treatment is more important than disease severity at diagnosis for prediction of early relapse in new-onset paediatric Crohn's disease. *Aliment Pharmacol Ther* 2018; **48**(11-12): 1242-50 AND Levine A, Chanchlani N, Hussey S, et al. Complicated Disease and Response to Initial Therapy Predicts Early

Surgery in Paediatric Crohn's Disease: Results From the Porto Group GROWTH Study. *Journal of Crohn's & colitis* 2020; **14**(1): 71-8.) Thus it was a pragmatic compromise with the timing of introduction of the maintenance treatment to give the induction treatment long enough to show its effect while recognising both treatments have a "lag period" of a few weeks before they become fully effective.

c) what are the traditional/accepted MTX levels in the blood? (as implicated on pg 21)

Response: There are no widely accepted levels for methotrexate in the literature at present. The inclusion of them in the study is exploratory.

A few comments:

Pg 7 - line 15-22: please revise to explain that biologics had previously (or traditionally) been reserved to those who failed immunomodulators (as, since the ECCO/ESPGHAN guidelines identified high risk features, many of us have been using biologics without previous IM if the patient presents with high risk features)

Response: Thanks. A sentence has been added to reflect this point.

"More recently in clinical practice patients deemed as high risk have been treated with a biologic without the use of a prior immunomodulator".

pg 7 - line 49-56: I don't understand what the sentence is saying; please modify sentences/explain further

Response: Thanks. The sentence

"There is a clear disparity between North America and Europe in terms of which form of immunosuppression first with both concerns about efficacy and safety lying behind these thus there is an urgent need for a head to head study in children to help inform the first choice of immunosuppression."

Has been modified to read

"There is a clear disparity between North America and Europe in terms of which form of immunosuppression is used initially with both concerns about efficacy and safety lying behind these differences, thus there is an urgent need for a head to head study in children to help inform the primary choice of immunosuppression."

Overall, please be careful of sentence structure and please add commas (as it otherwise makes it somewhat difficult to read)

Response: Thanks we have gone through the manuscript and adjusted the text where relevant based on this specific comment by the reviewer.

VERSION 2 – REVIEW

REVIEWER	Viktor Wintzell Karolinska Institutet, Sweden
REVIEW RETURNED	19-Mar-2020

GENERAL COMMENTS	Thanks for your responses and amendments. I have no further comments. Looking forward to seeing the results from this trial once published.
---